# First-Principle Calculation on Inelastic Electron Scattering in Diamond and Graphite

**DOI:** 10.3390/ma15093315

**Published:** 2022-05-05

**Authors:** Run-Qi Yan, Meng Cao, Yong-Dong Li

**Affiliations:** Key Laboratory of Physical Electronics and Devices of Ministry of Education, Faculty of Electronic and Information Engineering, Xi’an Jiaotong University, Xi’an 710049, China; yrq1996@stu.xjtu.edu.cn (R.-Q.Y.); mengcao@mail.xjtu.edu.cn (M.C.)

**Keywords:** diamond, graphite, secondary electron emission, inelastic scattering

## Abstract

In this work, we consider the inelastic scattering of incident electrons as a key process for analyzing the significant differences in secondary electron (SE) emission between diamond and graphite. Dielectric functions and energy- and momentum-dependent energy loss functions were obtained by first-principle calculations. These were then used to calculate the inelastic mean free path (IMFP) and stopping power in different directions. The results show that the properties of diamond are very close in different directions, and its IMFP is lower than that of graphite when the electron energy is higher than 30 eV. In graphite, the incident electrons may exhibit directional preferences in their motion. These results indicate that, in graphite, SEs are excited in deeper positions than in diamond, and more SEs move in a horizontal direction than in a vertical direction, which leads to the difference in secondary electron yield (SEY).

## 1. Introduction

Secondary electron (SE) emission from solid materials under electron bombardment plays an important role in many scientific and industrial applications. It is widely used in various fields of vacuum electronics, such as microscopic structure analysis, photoelectron spectroscopy, and SE spectroscopy. On the other hand, SE emission is undesired in many applications. For example, SE emission is the causative mechanism for a multipactor effect that damages high-power vacuum electron devices. Depending on the requirements, various surface treatments have been proposed to modify the secondary electron yield (SEY) of the material, among which, the coating of carbon-based materials is a widely used technique, especially using diamond and graphite.

Although diamond and graphite are both very good thermal conductors [1], their SEYs were found to be significantly different. The carbon films that show more graphite-like properties, such as amorphous carbon films, usually exhibit a lower SEY [2,3]. For amorphous carbon coatings, measurements with a SEY less than 1.2 were reported [4]. In contrast, the SE emission properties of diamond films with different treatments have been investigated, and the SEY of single crystal diamond was found to be greater than 10 [5,6]. The combined SEY and X-ray photoelectron spectroscopy study by Cimino et al. [7] showed that the formation of a graphitic film is the fingerprint and the actual chemical origin of the SEY reduction for a vast class of industrial materials. Allotropes of carbon, diamond, and graphite show great divergence in their SE emission properties, indicating that the internal structure of the materials has an important effect on SE emission. However, the theory about the connection between the internal structure of the material and the SE emission is still not well developed. Therefore, it is important to investigate the relationship between the structure of these two carbon materials and their SE emission properties in order to better understand SE emission.

Inelastic electron scattering is the fundamental physical mechanism associated with SE generation and has long attracted theoretical consideration. It occurs due to various interactions involving energy transfer between the primary incident electrons and the electrons inside the materials. The electrons are then excited in the material and generated as the SE. The probability of an electron being inelastically scattered is expressed in terms of the inelastic scattering cross section, often referred to as the inelastic mean free path (IMFP), which is defined as the average distance traveled by an energetic electron between successive inelastic scatterings. The IMFPs are calculated from energy- and momentum-dependent energy loss functions, the latter usually obtained by extrapolation of the optical energy loss function. Therefore, many scholars have intensively studied the methods for obtaining the energy loss function, which is a key step in the analysis of SE emission characteristics. Penn [8] developed an algorithm for optical energy loss function extrapolation and the evaluation of electron IMFPs based on dielectric function theory. This method was further developed by Ding and Shimizu [9,10,11] in a Monte Carlo study of SE emission. Ding’s group developed a consistent scheme on this method [12,13].

In recent years, the development of first-principle calculations has led to a method of calculating physical properties directly from basic physical quantities based on the principle of quantum mechanics. Time-dependent density functional theory calculations have been used to yield energy- and momentum-dependent energy loss functions and to evaluate the IMFP.

The results of density functional theory calculations can further describe SE generation and emission [14,15]. In this work, we have performed first-principle calculations for diamond and graphite and discussed the physical process of energy decay during the motion of incident electrons as they enter the interior of the material. Energy- and momentum-dependent energy loss functions were calculated for both materials, which were then used as input parameters for the evaluation of IMFP, with incident energies in the range of 0.1–1000 eV. By comparing the results of diamond and graphite, it is possible to reveal the essential reasons for the discrepancy of SE emission. The SE emission characteristics of materials can also be modified through structural modification.

## 2. Methods and Settings

Two types of structure were treated here, shown in Figure 1. The structure information of diamond and graphite crystals was obtained from the Material Project website [16]. The diamond structure was modeled using a two-atom FCC primitive unit cell (Fd$\bar{3}$m) with carbon atom sp${}^{3}$ hybridization. The lattice parameter *a* was 2.527 Å. The atoms of the graphite structure line up in the same plane and were modeled as a two-atom hexagonal primitive unit cell (P6/mmm) with carbon atom sp${}^{2}$ hybridization. The lattice parameters *a* and *b* were 2.467 Å, and *c* was 3.830 Å.

The first-principle calculation in this work was performed using a plane wave and a time-dependent density function perturbation theory module, with codes distributed with the QUANTUM ESPRESSO suite, version 6.6 [17,18,19]. The ultrasoft pseudopotential with the Perdew–Burke–Ernzerhof density function revised for solids [20,21] was used within a generalized gradient approximation [22,23] and a non-linear core correction, and scalar relativistic approximations were introduced. This ultrasoft pseudopotential has a relatively low cutoff radius, which is based on the fact that the bonds in the molecule are short, thus reducing computational time while ensuring the accuracy of the calculation [24].

Three steps were involved in obtaining the energy loss function data. First, relevant properties about the system, such as the ground state Kohn–Sham orbitals, the total energy, and charge density, were generated by the pw.x unit [17,18,19]. The turbo_eels.x unit [19,25,26,27] was then used to perform the Lanczos recursion at a given transferred momentum **q**. Finally, the charge density susceptibility was computed and output as a complex dielectric function and an energy loss function. This step was performed by the turbo_spectrum.x unit [17,18,19] as a post-processing step.

In the first step, a self-consistent calculation was performed to generate the ground-state plane wave, setting the convergence threshold of self-consistency to less than $10^{-5}$ Hartree, the mixing factor to 0.3, and the grid of k points in the first Brillouin zone to 15×15×15, with k being the wave vector. A higher convergence threshold and a lower mixing factor can facilitate the convergence of the self-consistent calculations. Since the ideal crystal structure is used in this work and the primitive cell itself is stable, lowering the convergence criterion and using a denser k-point grid can improve the computational efficiency.

For the time-dependent calculations, to obtain the energy loss function, the Liouville–Lanczos method was used, with the number of iterations set to 2000, along with the extrapolation technique. Crystallographic directions [100], [001], [011], and [111] were chosen for the momentum-dependence of the energy loss function calculation. At a small momentum transfer, there is a significant difference in the energy loss function, while the difference becomes smaller when the momentum transfer increases. The increments of momentum transfer are as follows: |**q**| = 0.01 between 0.005 and 1.755, |**q**| = 0.02 between 1.775 and 3.495, |**q**| = 0.05 between 3.555 and 5.255, |**q**| = 0.1 between 5.355 and 6.955, |**q**| takes the unit of 2π/a, and *a* is the lattice parameter.

For an electron with kinetic energy *E*, when transporting in a solid, inelastic scattering occurs, and energy is lost to the solid during transport. The expression of differential inverse IMFP is given by
(1)dλ−1dℏω=1πEa0∫q−q+dqqIm[−1ε(q,ω)],ℏq±=2m(E±E−ℏω)
where λ indicates the IMFP, *m* is the electron mass, $a_{0}$ is the Bohr radius, ℏq and ℏω are the momentum transfer and energy loss, and *ℏ* is the Planck constant. ε(q,ω) is the dielectric function, and the energy loss function is Im[−1/ε(q,ω)]. Fermi energy $E_{F}$ is the reference energy, and the energy loss dE on unit path length ds is given by the stopping power: (2)dEds=∫0E−EFℏωdω∫q−q+dqqIm[−1ε(q,ω)]

## 3. Results and Discussion

### 3.1. Dielectric Functions

Figure 2 demonstrates the complex dielectric functions in different crystallographic directions for a small momentum transfer at the optical limit (|q|→0). In this work, |q|≈0.01 Å${}^{-1}$ (|q|=0.005×2π/a), as in others works [27,28,29,30]. All energy values in this work are expressed with respect to the Fermi energy.

As a cubic structure with high symmetry, the dielectric functions of the diamond are equivalent along the crystallographic directions [001] and [011], and are close to that in direction [111], as shown in Figure 2a–c, respectively. The zero with a positive slope on the real part of dielectric function $\varepsilon_{1}$ is expressed as the character energy of self-sustaining oscillation, i.e., plasmon excitation [27]. It is shown that the characteristic energy of diamond associated with plasmon resonance is at approximately 29.4 eV. The plasmon-like peak on the energy loss function is located at approximately 35 eV, which is in agreement with the calculation by Timrov et al. [26] and Waidmann et al. [31] with the local-density approximation density function. The energy loss function curve of 0.01 Å${}^{-1}$ in the [001] direction in this work is smoother than the result of Timrov et al. for 0.15 Å${}^{-1}$ in the [100] direction (equivalent to [001]), which may be due to the interband transition caused by the local field effect of the density function in their study. Compared to the results of Waidmann et al. (0.5 Å${}^{-1}$), the energy loss function in this work shows anisotropy at smaller momentum transfers. This difference may be mainly attributed to the accuracy of the pseudopotential.

The dielectric functions and energy loss functions of graphite in Figure 2d–f exhibit completely different behaviors from those of diamond. The properties of the [001] direction perpendicular to the carbon atomic layer are significantly different from the properties of the other two in-plane directions due to the different symmetry. The real part of the dielectric function $\varepsilon_{1}$ is always greater than zero in the [001] direction, indicating that there is no plasmon excitation in this direction. There is no strong peak on the energy loss function in the [001] direction. For the other two in-plane directions, on the contrary, there are two zeros for $\varepsilon_{1}$ and two main peaks on the energy loss function. It can be speculated that these zeros are coupled with collective plasmon excitations [15], but have less association with interband transitions because of the irrelevance of the $\varepsilon_{2}$ peaks and energy loss function valleys [28]. Very high dielectric function values in the lower energy area are related to a high in-plane conductivity ($\varepsilon_{1}$) and the absorption of energy ($\varepsilon_{2}$).

Energy loss functions with a greater momentum transfer were also calculated. The momentum-dependent energy loss function can be regarded as the susceptibility of a system to a single- or bulk-plasmon excitation at a given energy and momentum transfer [32]. Figure 3 shows the energy- and momentum-dependent energy loss functions of diamond and graphite. |q| ranges from 0.005 × 2π/a to 0.405 × 2π/a, with steps of 0.02 × 2π/a.

For the three directions of [100], [011], and [111], the energy loss functions of diamond have a similar trend. On the curve with the smallest |**q**|, there is a plasmon peak centered at around 34 eV. As |**q**| increases, another peak at around 23 eV progressively becomes more pronounced. Both peaks become flatter on the lower energy side and steeper on the high energy side. The peak at around 34 eV decreases more rapidly than the peak at around 27 eV, with decrements of 3.5% and 1.7%, respectively. The difference between these results and Waidmann et al.’s [31] results is the peak near 23 eV. In this work, the peak near 23 eV becomes apparent when the momentum transfer increases to 1.00 Å${}^{-1}$. According to Waidmann et al., this peak appears at a momentum transfer of 0 and becomes less pronounced when the momentum transfer increases, especially in the [001] direction.

The energy loss functions of graphite are more complex than those of diamond. Unlike the diamond, the energy loss functions of graphite are various in different directions. In the [100] direction parallel to the atomic plane, there is a strong peak centered around 30 eV and a lower peak around 8 eV. As |**q**| increases, the intensity of the two peaks decrease by 48.0% and 41.8%, respectively. In the [001] direction, the distribution of the peaks is clearly different. There are four ripples in the energy range from 10 to 50 eV, and their spacing varies little with **q** (less than 10%). These peaks have very little variation in the position of their maximum at high |**q**| values and correspond to interband transition or core excitation [31]. In addition, the energy loss function is also given along the [111] direction, which has an angle to the atomic layer. The distribution of and variation in energy loss function peaks along [111] are similar to those along [100], whereas the intensity of the energy loss function peaks is lower. The [111] direction can be regarded as both parallel and perpendicular to the atomic plane, whereas the energy loss function is closer to the [100] direction, indicating that the energy loss function distribution in the [100] direction plays a major role in the overall energy loss of electrons.

### 3.2. Transport of Energetic Electrons

Based on the energy loss functions, we calculated the IMFPs using Equation (Equation 1) and shown in Figure 4. Furthermore, stopping powers are also given in Figure 5 using Equation (Equation 2). In the discussion, the data are compared with the results from reference [33,34]. The IMFP reference data for both diamond and graphite above 100 eV are close to those for graphite in the [100] direction. The reference values are approximately 90% of the graphite [100] value at 100 eV, decreasing with increasing energy to approximately 75% at 1000 eV. For the stopping power, all peaks occur in relatively close proximity. The peak of the reference data is between diamond [111] and graphite [100], the peak of diamond [111] is 24.10% higher than its reference, the graphite reference is 26.25% higher than the peak of graphite [100], and the diamond reference is 31.3% higher than the graphite reference.

For the diamond, the energy loss functions are similar for all directions, and the results for any direction are representative of all directions. Therefore, only the [111] direction in diamond and the two unrelated directions in graphite are given. The curve of the IMFP vs. the energy has a universal shape [35]. It is V-shaped: first decreasing and then increasing, with a minimum around 78 eV. The distribution of stopping power is opposite that of IMFP. The smaller the IMFP, the greater the probability of inelastic scattering and electron energy loss over the same moving distance, and the bigger the corresponding stopping power.

There are evident differences in the IMFPs of graphite between the in-plane direction [100] and the perpendicular direction [001]. In general, the IMFP in [001] is larger than that in [100]. For an energy less than 30 eV, the former is one order of magnitude larger than that in the [100] direction. Their minimum values are both around 80 eV, approximately equal to 4.8 Å and 6.7 Å, respectively. The stopping power values of graphite also present great differences in different directions. Values in the [001] direction are lower than those in the [100] direction.

Considering the difference in the IMFP between the reference data and the results in the present work, diamond and graphite belong to different cases. The results of Tanuma et al. [33,34] are derived from the optical energy loss function to a finite momentum transfer. Moreover, the shape and value of the energy loss function are constrained to decrease monotonically as the momentum transfer increases, and no new peaks appear [12]. In addition, for diamond and graphite, the IMFPs calculated from the optical data have relatively high uncertainty [35]. Therefore, the reference data for diamonds may overestimate the IMFP and underestimate the stopping power for the reasons mentioned above. For graphite, the IMFP in the [001] direction is higher than that in the [100] direction because of the lower energy loss function according to the inversion relation in Equation (Equation 1). The energy loss function in the [100] direction plays a major role and is compared with the reference data. As the first few curves in Figure 3d show, the IMFP in [100] is higher than the reference data, probably because the energy loss function in the small momentum transfer region decays faster than that in the high momentum transfer region, whereas the extrapolation in the reference data may decay more slowly and produce a larger total value of the energy loss function.

### 3.3. Secondary Electron Excitation

We present here a brief discussion on the SE excitation in the material based on the obtained calculations. According to Equation (Equation 1), the larger energy loss function of |**q**| also contributes to the inelastic cross section, whereas the plasmon excitation is expected to have little effect [31]. Due to the similar properties in all three directions, the movement of the energetic electron in the diamond can be considered the same in all directions. In this case, a large amount of internal SE can be produced in the diamond in all directions by interband transitions and core excitation. For graphite, the in-plane direction in which SEs are mainly generated plays a major role in the energy loss function. At the same time, the energy loss function associated with the [001] direction, which is perpendicular to the atomic plane, is relatively small, resulting in only a small amount of internal SEs being produced when the incident electrons move in this direction.

After being generated inside the material, SE can only be detected after escaping from the surface. That is, the SE must undergo a movement in the direction perpendicular to the surface. The IMFP and stopping power are used here to explain the difference in the SEY between diamond and graphite.

Considering the energy range of a few tens of volts, the probability of SE excitation by incident electrons is relatively high. In this energy interval, the minimum value of the diamond IMFP is smaller than that of the graphite, implying that SE is generated at a shallower location under the material surface. Moreover, the stopping power of the diamond is larger than that of the graphite, resulting in more internal SEs being excited by the incident high-energy electrons. According to the energy spectrum of SE, most of the SEs are within a few electron volts of energy. The diamond has a larger IMFP in the interval of a few electron volts, indicating that most SEs can travel further in the material, leading to a high probability of excitation. In conclusion, more SEs are produced near the surface, and the excitation properties are similar in all directions, which may be a factor related to the high SEY in diamond. On the other hand, the stopping power value of graphite is relatively small, and fewer internal SEs are excited by the incident electrons in this energy range. In addition, the band structure of graphite leads to less SE production in the lower energy range [36]. In graphite, the IMFP in the [001] direction is higher than that in the [100] direction, whereas the stopping power in the [001] direction is smaller than that in the [100] direction, which implies a preference for the [001] direction during the motion of the incident electrons. The higher IMFP in the [001] direction compared to diamond suggests that SEs are produced in the material more deeply in graphite than in diamond. Therefore, more SEs move within the atomic plane rather than in the perpendicular direction, and SEs in the plane are more difficult to escape, resulting in a very low SEY in the graphite structure.

## 4. Conclusions

This work discusses the inelastic scattering process of diamond and graphite and attempts to explain the differences between their SE emission. In diamond, 3D symmetric sp${}^{3}$ hybridization leads to quasi-isotropy in a momentum-dependent energy loss function and the IMFP, which means that the pattern of SE generation and electron motion in all directions within the material is very close. Graphite, on the other hand, shows different properties in different directions due to its laminar structure. Incident electrons tend to move deeper into the material, and the SE inside the graphite is biased to move in the horizontal direction. The energetic electrons inside the material, whether incident electrons or SEs generated within the material, may need to undergo multiple inelastic scattering towards the surface before they are emitted from inside the material. In comparison, energetic electrons in diamond are more likely to be produced near and move toward the surface than in graphite. This discrepancy may lead to the small SEY of graphite.

This paper does not consider the differences in elastic scattering, which is another part of electron scattering. Monte Carlo simulation, a well-established method based on elastic scattering properties, can be used to gain a detailed understanding and simulation of the SE emission. In addition, other factors, such as material surface potential and surface morphology, also have an effect on SE emission. More research work is needed for a clear relationship between different factors and SE emission and the degree of influence. 

## Figures and Tables

**Figure 1 materials-15-03315-f001:**
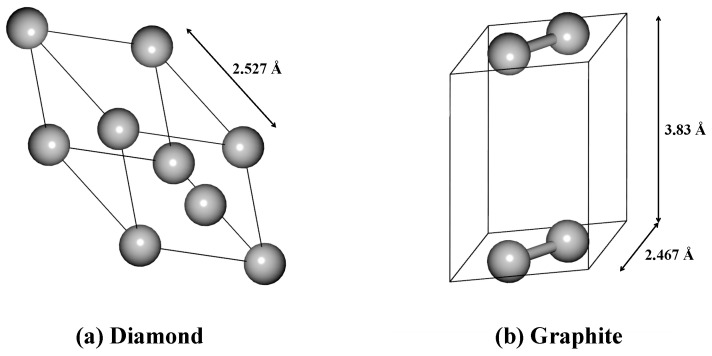
Structures for the primitive cells of diamond and graphite. The solid line is the boundary of the primitive cells, and the atoms on the boundary are equivalent.

**Figure 2 materials-15-03315-f002:**
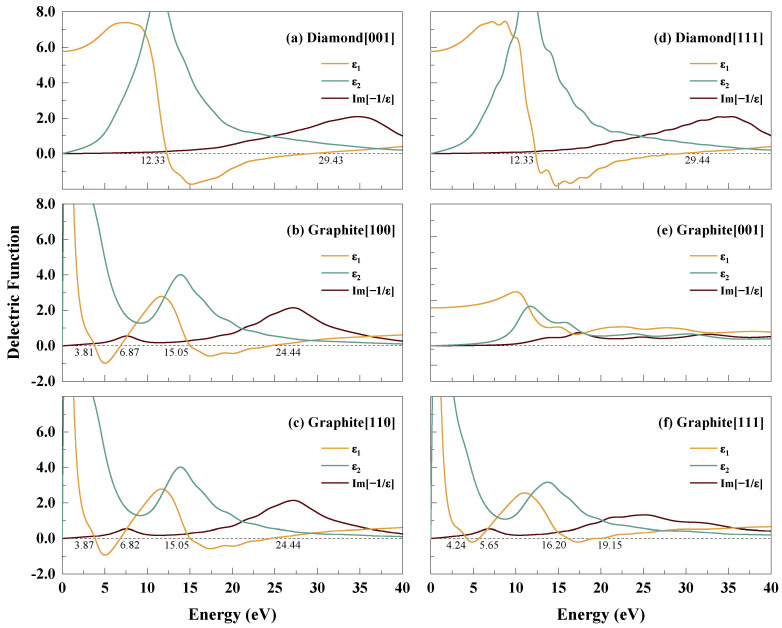
Dielectric functions and energy loss functions at the optical limit for the [001], [011], and [111] directions of diamond and the [100], [100], and [111] directions of graphite.

**Figure 3 materials-15-03315-f003:**
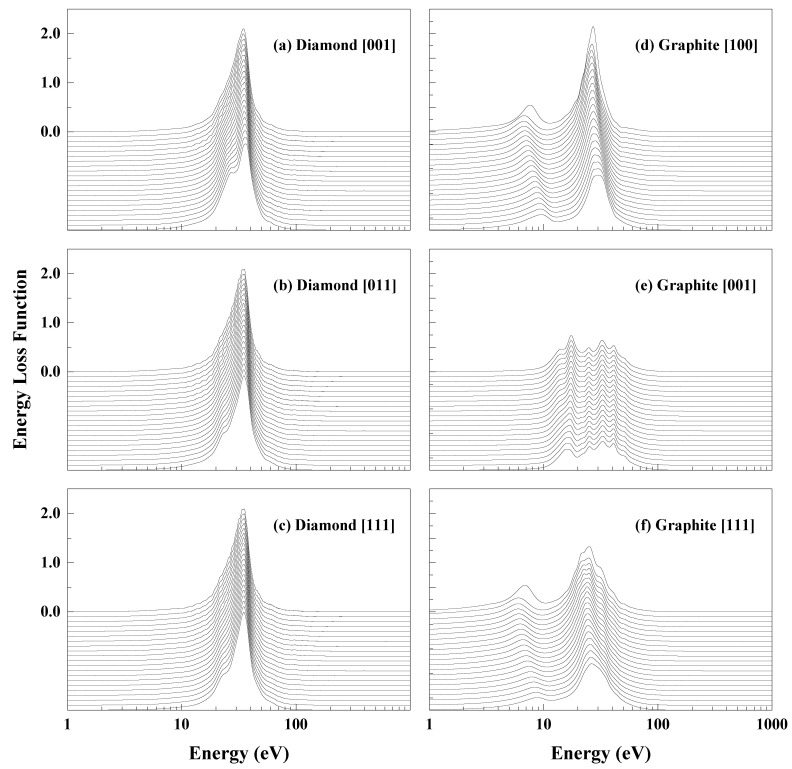
**q**-dependent energy loss functions for the [001], [011], and [111] directions of diamond and the [100], [100], and [111] directions of graphite. For clarity, zero on energy loss axis coincides with the energy loss function calculated for |**q**|≈0.01 Å${}^{-1}$ (0.005 × 2π/a). Successive energy loss functions for increasing q-values have been moved downward to |**q**|≈1.00 Å${}^{-1}$ (0.405 × 2π/a) in decrements of 0.02 × 2π/a.

**Figure 4 materials-15-03315-f004:**
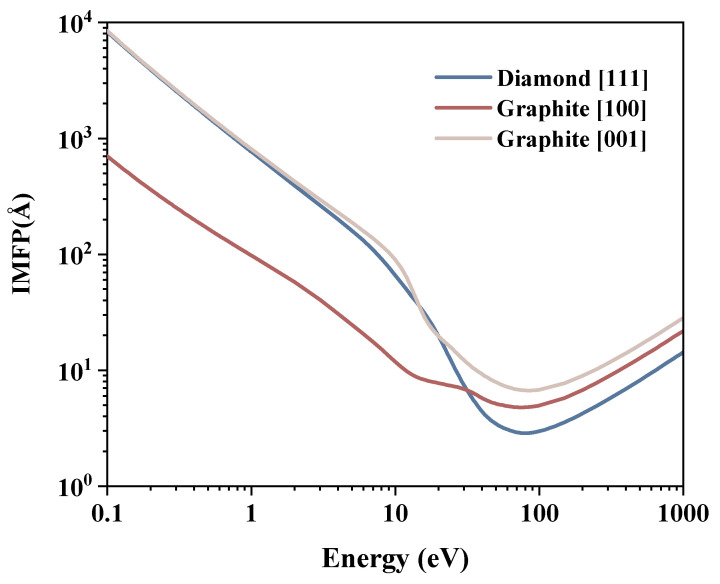
IMFP of diamond in the [111] direction and of graphite in the [100] and [001] directions.

**Figure 5 materials-15-03315-f005:**
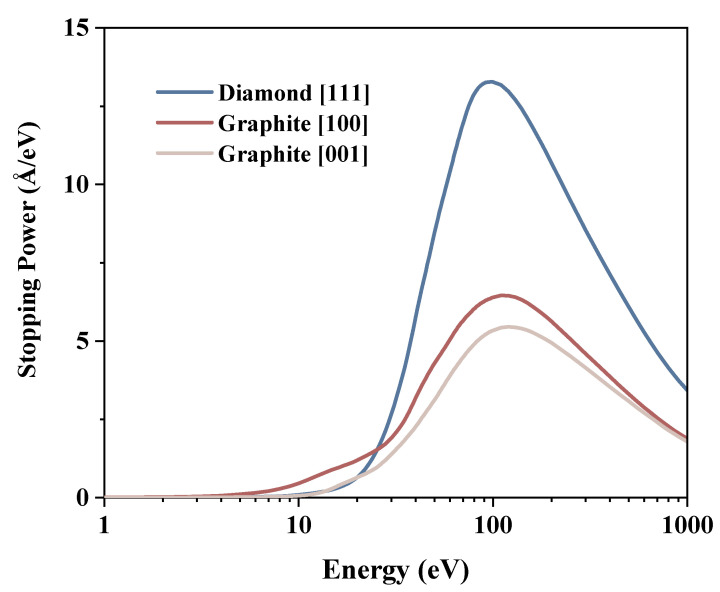
Stopping power for the [111] direction of diamonds and the [100] and [001] directions of graphite.

## Data Availability

The data are contained within the article.

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
