# Peer review of "First-Principle Calculation on Inelastic Electron Scattering in Diamond and Graphite"

_materials, 2022, doi:10.3390/ma15093315_

Round 1
Reviewer 1 Report
The authors of this study have focused on the electron emission properties of diamond and graphite, in various directions.
- The ms is not well written, unclear and hence it needs significant improvement before its publication in anywhere else. For instance, the following sentence repeated both in the abstract and conclusion section of the ms is incomplete, and it does not make any sense. “Whereas, there are obvious directional dependence in graphite.”
- The authors must provide more clear pictures, explanation of the geometry and lattice properties, which are lacking in the current version of the ms.
- The abstract and conclusion of the paper reads almost the same. Both parts should be revised appropriately.
- The manuscript lacks discussion on the details of calculations and protocols of QE used.
- 1 Atomic structure of diamond and graphite. Is it the atomic structure, or the electronic structure? Do they correspond to unit-cell, primitive-cell, or conventional cell?
- Background references related to the study are not appropriately cited, or discussed briefly.
- Discussion of basic sciences should be improved significantly.
Author Response
Response to Reviewer 1 Comments
Point 1: The ms is not well written, unclear and hence it needs significant improvement before its publication in anywhere else. For instance, the following sentence repeated both in the abstract and conclusion section of the ms is incomplete, and it does not make any sense. “Whereas, there are obvious directional dependence in graphite.”
Response 1: We checked the ms and submit language editing. The mentioned sentence is revised as “In graphite, the incident electrons may exhibit directional preferences in their motion.”
Point 2: The authors must provide more clear pictures, explanation of the geometry and lattice properties, which are lacking in the current version of the ms.
Response 2: The structure of the materials are described in details in the modified version. A picture of primitive cell used in calculation was replaced for the structure of Figure 1.
Point 3: The abstract and conclusion of the paper reads almost the same. Both parts should be revised appropriately.
Response 3: The abstract is revised to focus on the process of this researching. And the conclusion is revised to focus on the patterns found in the calculation and discussion.
Point 4: The manuscript lacks discussion on the details of calculations and protocols of QE used.
Response 4: The intention of the parameter settings calculated by QE has been added in the revised version at the end of the second and fourth paragraph of section 2, to explain the reasonableness of the parameter settings. The added related description are “This ultrasoft pseudopotential has a relatively low cutoff radius, which is based on the fact that the bonds in the molecule are short, thus reducing computational time while ensuring the accuracy of the calculation (Corso, 2014).”, and ”A higher convergence threshold and a lower mixing factor can facilitate the convergence of the self-consistent calculations. Since the ideal crystal structure is used in this work and the primitive cell itself is stable, lowering the convergence criterion and using a denser k-point grid can improve the computational efficiency.”, respectivrly.
Point 5: Atomic structure of diamond and graphite. Is it the atomic structure, or the electronic structure? Do they correspond to unit-cell, primitive-cell, or conventional cell?
Response 5: It is the atomic structure correspond to primitive-cell. The primitive-cell of diamond and graphite includes two atoms and related to FCC and hexagonal crystal system, respectively. We revised the related description at the end of the first paragraph in section 2 as “The diamond structure was modeled using a two-atom FCC primitive unit cell (Fd3(—)m) with carbon atom sp3 hybridization. The lattice parameter a is 2.527Å. The atoms of the graphite structure line up in the same plane and are modeled as a two-atom hexagonal primitive unit cell (P6/mmm) with carbon atom sp2 hybridization. The lattice parameters a and b are 2.467Å, and c is 3.830Å.”
Point 6: Background references related to the study are not appropriately cited, or discussed briefly.
Response 6: We have reorganized the physical background of inelastic scattering to give a clear description. The relationship between the background and the references has also been logically revised. We revised the related description at the end of the third paragraph in section 1 as “The probability of an electron being inelastically scattered is expressed in terms of the inelastic scattering cross section, often referred to as the inelastic mean free path (IMFP), which is defined as the average distance traveled by an energetic electron between successive inelastic scatterings. The IMFPs are calculated from energy- and momentum-dependent energy loss functions, the latter usually obtained by extrapolation of the optical energy loss function. Therefore, many scholars have intensively studied the methods for obtaining the energy loss function, which is a key step in the analysis of SE emission characteristics. Penn (Penn, 1987) developed an algorithm for optical energy loss function extrapolation and the evaluation of electron IMFPs based on dielectric function theory. This method was further developed by Ding and Shimizu (Ding, 2004; Ding, 2001; Ding, Shimizu, 1996;) in a Monte Carlo study of SE emission. Ding's group developed a consistent scheme on this method (Mao, 2008; Da, 2014). ”
Point 7: Discussion of basic sciences should be improved significantly.
Response 7: A third subsection has been added in Section 3 to discuss the physical processes and their differences in the two materials where the incident electrons move through the material and excite the secondary electrons in the internal material.
Reviewer 2 Report
The authors use the calculation of the first principle to obtain the inelastic dispersion of diamond and graphite using free software.
They clearly show their results but there is a lack in the discussion.
In sections 3.1 and 3.2 the authors cannot only show the results but also argue their results with those obtained by other authors.
It comes to seem like a technical report with no discussion. Authors should check if their work would be reproducible with the data and parameters shown in the text.
Author Response
Response to Reviewer 2 Comments
Point 1: They clearly show their results but there is a lack in the discussion.
Response 1: A third subsection is added in Section 3 to discuss the physical processes. We compared the IMFP and stopping power values for diamond and graphite in the same energy range and, accordingly, qualitatively discuss the differences in the physical processes of the incident electrons moving in the two materials and then propose an explanation for the differences in the SEY of the two materials.
Point 2: In sections 3.1 and 3.2 the authors cannot only show the results but also argue their results with those obtained by other authors.
Response 2: The discussion of results in this work and references was added in the mentioned sections. We proposed an explanation for the differences. We added the related description at the last paragraph in section 3.2 as “Considering the difference in the IMFP between the reference data and the results in the present work, diamond and graphite belong to different cases. The results of Tanuma et al. (Tanuma, 2005, 2008) are derived from the optical energy loss function to a finite momentum transfer. Moreover, the shape and value of the energy loss function are constrained to decrease monotonically as the momentum transfer increases, and no new peaks appear (Mao 2008). In addition, for diamond and graphite, the IMFPs calculated from the optical data have relatively high uncertainty (Szajman, 1981). Therefore, the reference data for diamonds may overestimate the IMFP and underestimate the stopping power for the reasons mentioned above. For graphite, the IMFP in the [001] direction is higher than that in the [100] direction because of the lower energy loss function according to the inversion relation in Equation (1). The energy loss function in the [100] direction plays a major role and is compared with the reference data. As the first few curves in Figure 3(d) show, the IMFP in [100] is higher than the reference data, probably because the energy loss function in the small momentum transfer region decays faster than that in the high momentum transfer region, while the extrapolation in the reference data may decay more slowly and produce a larger total value of the energy loss function.”
Point 3: It comes to seem like a technical report with no discussion. Authors should check if their work would be reproducible with the data and parameters shown in the text.
Response 3: The intention of the parameter settings calculated by QE has been added in the revised version at the end of the second and fourth paragraph of section 2, to explain the reasonableness of the parameter settings. The added related description are “This ultrasoft pseudopotential has a relatively low cutoff radius, which is based on the fact that the bonds in the molecule are short, thus reducing computational time while ensuring the accuracy of the calculation (Corso, 2014).”, and ”A higher convergence threshold and a lower mixing factor can facilitate the convergence of the self-consistent calculations. Since the ideal crystal structure is used in this work and the primitive cell itself is stable, lowering the convergence criterion and using a denser k-point grid can improve the computational efficiency.”, respectivrly.
Round 2
Reviewer 1 Report
The authors have considered most of my comments and revised their manuscript. But the conclusion is not rewritten. Moreover, I can not see the certificate from the english editing agency. So, I still believe the ms requires an extensive English language editing.
Author Response
Response to Reviewer 1 Comments
Point 1: The conclusion is not rewritten.
Response 1: The conclusion is rewritten as "This work discusses the inelastic scattering process of diamond and graphite and attempts to explain the differences between their SE emission. In diamond, 3D symmetric sp3 hybridization leads to quasi-isotropy in a momentum-dependent energy loss function and the IMFP, which means the pattern of secondary electron generation and electron motion in all directions within the material is very close. Graphite, on the other hand, shows different properties in different directions due to its laminar structure. Incident electrons tend to move deeper into the material, and the SE inside the graphite is biased to move in the horizontal direction.
The energetic electrons inside the material, whether incident electrons or secondary electrons generated within the material, may need to undergo multiple inelastic scattering towards the surface before they are emitted from inside the material. In comparison, Energetic electrons in diamond are more likely to be produced near and move toward the surface than in graphite.
This discrepancy may lead to the small SEY of graphite.
This paper does not consider the differences in elastic scattering, which is another part of electron scattering. Monte Carlo simulation, a well-established method based on elastic scattering properties, can be used to gain a detailed understanding and simulation of the SE emission. In addition, other factors such as material surface potential and surface morphology also have an effect on SE emission. More research work is needed for a clear relationship between different factors and SE emission and the degree of influence."
Point 2: Lack of the certificate from the english editing agency.
Response 2: I did not find where should submit the certificate. The editor replied to me that I could send them the edited English certificate via email. They will help upload this file into the submission system. And I have sent.
